# Complex Cell Type-Specific Roles of Autophagy in Liver Fibrosis and Cirrhosis

**DOI:** 10.3390/pathogens9030225

**Published:** 2020-03-18

**Authors:** Tzu-Min Hung, Chih-Chiang Hsiao, Chih-Wen Lin, Po-Huang Lee

**Affiliations:** 1Department of Surgery, National Taiwan University Hospital and National Taiwan University College of Medicine, Taipei 100225, Taiwan; mean6722@ms65.hinet.net (T.-M.H.); ansonsfaith@gmail.com (C.-C.H.); 2Department of Medical Research, E-DA Hospital, Kaohsiung 824410, Taiwan; 3Department of Surgery, E-Da Hospital, Kaohsiung 824410, Taiwan; 4Division of Gastroenterology and Hepatology, Department of Medicine, E-Da Hospital and I-Shou University, Kaohsiung 824410, Taiwan; lincw66@gmail.com; 5School of Medicine, College of Medicine, I-Shou University, Kaohsiung 840301, Taiwan

**Keywords:** autophagy, liver cell, fibrosis, cirrhosis

## Abstract

The lysosomal degradation pathway, or autophagy, plays a fundamental role in cellular, tissue, and organismal homeostasis. A correlation between dysregulated autophagy and liver fibrosis (including end-stage disease, cirrhosis) is well-established. However, both the up and downregulation of autophagy have been implicated in fibrogenesis. For example, the inhibition of autophagy in hepatocytes and macrophages can enhance liver fibrosis, whereas autophagic activity in hepatic stellate cells and reactive ductular cells is permissive towards fibrogenesis. In this review, the contributions of specific cell types to liver fibrosis as well as the mechanisms underlying the effects of autophagy are summarized. In view of the functional effects of multiple cell types on the complex process of hepatic fibrogenesis, integrated approaches that consider the role of autophagy in each liver cell type should be a focus of future research.

## 1. Introduction

Mounting evidence indicates that alterations in the autophagic process drive the progression of various liver diseases, including α1-antitrypsin deficiency [1,2,3], alcoholic and nonalcoholic fatty liver diseases [4,5,6], viral hepatitis B [7,8], viral hepatitis C [9,10], and hepatocellular carcinoma [8,11,12]. Our understanding of the roles of autophagy in the control of liver fibrosis, a precursor for liver cirrhosis, is improving. However, the precise effects of autophagy in liver fibrosis vary widely, with opposing functions depending on the cell type [13]. Most review articles have broadly focused on the role of autophagy in liver diseases, with limited information on cell type-specific roles of liver fibrosis [13,14,15]. Therefore, this review summarizes emerging information about the role of autophagy in each resident liver cell involved in the development of fibrosis (Figure 1). After briefly reviewing autophagy-related processes in various cell types involved in fibrogenesis, potential strategies for therapeutic modulation are discussed.

## 2. Autophagy

Autophagy, literally meaning “self-eating,” is an intracellular degradation process by which protein aggregates, damaged organelles, and invading microbes are delivered to lysosomes to maintain cellular homeostasis [16]. Autophagy is initiated by the formation of double-membrane vesicles, called autophagosomes, which fuse with lysosomes to form autolysosomes for subsequent degradation. Autophagy execution involves a collection of evolutionarily conserved gene products, known as the Atg proteins, that are required to form the isolation membrane and the autophagosome [17,18]. The autophagosome formation includes two major steps: nucleation and elongation of the isolation membrane. For the nucleation step, the Atg1/ULK kinase complex, the class III phosphatidylinositol 3 (PI3) kinase complex, and PI3P effectors and their associated proteins are essential, whereas the Atg12- and Atg8-conjugation systems are critical for the elongation step [17,18]. 

Microtubule-associated protein 1 light chain 3 (LC3), a homologue of yeast Atg8, localizes to autophagosomal membranes after post-translational modifications [19]. The C-terminal fragment of LC3 is cleaved immediately following synthesis to yield a cytosolic form called LC3-I. A subpopulation of LC3-I is activated by Atg7, transferred to Atg3, and finally covalently linked to phosphatidylethanolamine and converted to an autophagosome-associating form, LC3-II [19]. The amount of LC3-II correlates well with the number of autophagosomes [20]. Conversion of LC3-I to LC3-II is greatly enhanced by culturing cells under starvation conditions that efficiently induce autophagy. This characteristic LC3 conversion can be used to monitor autophagic activity [20]. The human LC3 gene family has three members namely LC3A, LC3B, and LC3C. LC3B is most commonly used for autophagy assays [17].

An alternative method for detecting the autophagic flux is measuring p62/SQSTM1 degradation. p62/SQSTM1 is selectively incorporated into autophagosomes through direct binding to LC3 and is efficiently degraded by autophagy [21]; thus, the total cellular expression levels of p62/SQSTM1 correlate inversely with autophagic activity. For example, in autophagy-deficient cells, a starvation-induced reduction in p62/SQSTM1 levels is not observed, and instead p62/SQSTM1 accumulates [22].

Autophagy was originally discovered in studies of liver tissues, in which it was found to maintain energy balance [23]. In the liver, autophagic activity is not only essential for replenishing the free amino acid pool by protein breakdown but also contributes to the mobilization and hydrolysis of lipid stores and glycogen, thereby contributing substantially to the maintenance of cellular energy [24]. In light of the multiple mechanisms by which autophagy participates in the control of liver metabolism, it is no surprise that the dysregulation of autophagy has been implicated in many liver diseases [25].

## 3. Cellular and Molecular Determinants of Liver Fibrosis 

Liver fibrogenesis is a dynamic and highly integrated molecular, tissue, and cellular process that drives the progression of chronic liver diseases towards liver cirrhosis and hepatic failure [26]. The majority of research has focused on the responses of hepatic stellate cells (HSCs) and myofibroblasts, given their critical roles in extracellular matrix production [27]. However, liver injury elicits a complex multicellular response involving other resident cells, including hepatocytes, macrophages, liver sinusoidal endothelial cells (LSECs), and distinct families of infiltrating immune cells. 

It is also important to note the role of epithelial components in liver fibrosis via the so-called ductular reaction (DR) [28]. The DR is a common response to injury observed in a variety of liver diseases [29]. It is characterized by the appearance of reactive ductular cells (RDCs), which are a population of cholangiocyte-like epithelial cells able to orchestrate a complex mixture of the extracellular matrix, mesenchymal cells, and inflammatory infiltrate [30]. Accumulating evidence indicates that DR is closely correlated with the severity of fibrosis in several human liver diseases [31,32,33]. Therefore, DR is gaining interest as a therapeutic target for liver fibrosis.

We outline the roles of autophagy in various cells involved in fibrogenesis, including HSCs, RDCs, hepatocytes, cholangiocytes, macrophages, and LSECs.

### 3.1. HSCs

In most liver diseases, autophagy is mainly protective; for example, it promotes the degradation of lipid droplets in fatty acid disease and protein aggregates in alcoholic liver disease [4,5]. Therefore, evidence suggesting that autophagy is a target for the prevention of HSC activation conflicts with the conventional viewpoint. Two independent groups have reported that autophagy can regulate lipid droplets and then drive the activation of HSC [34,35]. HSC activation, both *in vitro* and in rodent models of liver injury, is associated with features of autophagy induction, including a marked increase in autophagic vacuoles, LC3-II levels, and autophagic flux [34,35]. The blocking of autophagy in cultured cells, with 3-methyladenine (3-MA) or specific siRNAs targeting *Atg5* or *Atg7*, leads to the attenuation of HSC activation and fibrogenesis [34]. Using a mouse strain with the HSC-specific deletion of *Atg7*, Hernández-Gea et al. observed the attenuation of fibrosis after liver injury by either carbon tetrachloride (CCL_4_) or thioacetamide [34]. While these two studies provide clear evidence for the role of autophagy in promoting liver fibrosis, there is insufficient clinicopathologic evidence supporting this link. 

### 3.2. Reactive Ductular Cells

Autophagy has not been measured directly in the liver of patients with cirrhosis; accordingly, we performed an immunodetection assay of LC3B puncta in fresh-frozen tissues. A dual-immunofluorescence analysis of cirrhotic livers demonstrated the localization of LC3B to CK19-positive RDCs [36], providing the first *in vivo* evidence, to our knowledge, for the regulation of autophagy in fibrogenic cells other than HSCs. A progressive increase in LC3B and CK19 expression was observed in diseased human liver as the severity of fibrosis increased from the F1 stage to the F2 and F4 stages [36]. Treatment of rat AAF/CCL_4_ fibrotic models with chloroquine, which blocks autophagic degradation in the lysosome, decreased the expression of CK19 and profibrogenic targets (COL1A1, α-SMA, and TGF-β), and blunted liver fibrosis. These findings suggest that autophagy promotes DR expansion and fibrogenesis. 

DR is a process that occurs hand in hand with HSC activation in a subset of liver diseases [28]. RDCs are able to secrete multiple soluble pro-fibrogenic factors acting on HSCs and myofibroblasts [37]. In this regard, work from our laboratory demonstrated the enrichment of α-SMA-positive cells in close proximity to LC3B-positive ductule structures (observed both in the livers of cirrhotic patients and in a rat model of AAF/CCL_4_-induced liver cirrhosis), providing evidence that autophagy may have a role in this paracrine interaction between RDCs and myofibroblasts.

To further investigate the effect of autophagy on specific RDCs of interest, we used EpCAM-positive selection to purify RDCs from a rat model of AAF/CCL_4_-induced liver cirrhosis [38]. RDCs from AAF/CCL_4_ livers exhibited higher levels of autophagy than those of normal livers. These cells showed morphological and functional characteristics of mesenchymal cells. Blocking autophagy using bafilomycin A1 or siRNA targeting *Atg7* reduced the expression of mesenchymal markers in these RDCs from AAF/CCL_4_ livers, indicating that autophagy is involved in the regulation of the mesenchymal phenotype of RDCs. Importantly, in cirrhotic human livers, RDCs positive for LC3B also showed increased expression of TGF-β and fibroblast specific protein-1 [38]. Our data suggest that autophagy is activated in RDCs and is required for the mesenchymal transition during the DR, processes that are critically involved in the pathogenesis of cirrhosis.

### 3.3. Hepatocytes

Recurrent epithelial injury is a prominent driving factor in the pathogenesis of progressive fibrosis [39]. Autophagy in hepatocytes is generally considered a protective pathway [40,41]. Therefore, hepatocyte autophagy may protect against fibrosis. Ni et al. reported that the hepatocyte-specific deletion of *Atg5* results in increased apoptosis, inflammation, and fibrosis in mouse liver, which is mediated by the accumulation of p62/SQSTM1 and the activation of Nrf2 [42]. Furthermore, the persistent activation of Nrf2 contributes to Atg5-deficiency induced liver injury by enhancing aberrant protein accumulation and disrupting the homeostasis of pro- and anti-apoptotic proteins [42].

The immunohistochemical detection of autophagy markers (e.g., LC3 and p62/SQSTM1) is a common method for investigating autophagy process in human tissues [43]. In a theoretical model, elevated LC3 levels should indicate increased autophagosome formation. The analysis of p62/SQSTM1, a reliable method to monitor autophagy, may be of further value in this context [43]. In support of Ni et al., who found that defective hepatic autophagy can lead to liver fibrosis, our group examined the immunohistochemical expression of LC3B and p62/SQSTM1 in livers taken from patients with cirrhosis. Unlike the negative control in cirrhosis and/or the non-cirrhotic liver, hepatocytes in the cirrhotic liver showed the co-expression of LC3B and p62/SQSTM1, suggesting impaired hepatocyte autophagy [38]. In contrast, the DR in cirrhotic livers showed strong expression of LC3B but no expression of p62/SQSTM1, indicating the induction of autophagy [38]. Based on the different structures showing different reactivity on the same slide, we concluded that the autophagy status differs between RDCs and hepatocytes in cirrhotic livers.

### 3.4. Cholangiocytes

Cellular senescence of cholangiocytes and a senescence-associated secretory phenotype lead to the production of proinflammatory cytokines, thereby modifying the milieu of the bile duct and triggering a fibro-inflammatory response in primary biliary cirrhosis (PBC) [44]. 

Sasaki et al. reported that autophagy is specifically upregulated in cholangiocytes along with senescence in PBC [45]. Vesicles positive for the autophagy marker LC3 accumulate in the cytoplasm of damaged bile ducts expressing senescence markers in the livers of PBC patients [45]; this suggests that autophagy could induce and facilitate cholangiocyte senescence. Furthermore, the aggregation of p62/SQSTM1, an adaptor molecule that is selectively degraded via autophagy, also increases in damaged bile ducts in the livers of PBC patients [46]. Based on both the accumulation of LC3-positive vesicles and the aggregation of p62/SQSTM1, a decrease in autophagic flux and dysfunctional autophagy in cholangiocytes may contribute to PBC. 

### 3.5. Macrophages

Hepatic inflammation contributes to progressive liver injury and fibrosis. Among the various immune cells, macrophages are the key components of innate immune response in the liver [47]. In chronic liver injury, macrophages can release mediators that promote hepatocyte apoptosis and contribute to inflammatory cell recruitment and activation of hepatic fibrogenic cells. Emerging evidence suggests that autophagy in macrophages regulates the inflammasome and protects against liver injury [48]. Liu et al. reported that macrophages were polarized into a more pro-inflammatory M1 phenotype when macrophage-specific Atg5 knockout mice were challenged with a high fat diet [49]. Lodder et al. reported that autophagy-deficient macrophages enhance the fibrogenic properties of hepatic myofibroblasts via the secretion of the inflammatory cytokines IL-1A and IL-1B [50]. Moreover, mice with an *Atg5* deficiency in the myeloid lineage show enhanced hepatocyte apoptosis, inflammatory cell recruitment, and are more susceptible to liver fibrosis [50]. Based on these findings, macrophage autophagy is an anti-inflammatory pathway in the liver, with hepatoprotective effects and antifibrogenic effects via paracrine interactions with hepatic myofibroblasts.

### 3.6. Liver Sinusoidal Endothelial Cells 

LSECs constitute the liver’s first line of defense owing to their unique position of lining the sinusoidal lumen. LSECs are characterized by the possession of cellular pores or fenestrae, which are critical for the maintenance of homeostasis in the liver parenchyma. In pathological conditions, LSECs lose their protective properties and exhibit capillarization (defenestration or loss of fenestrae), and these cells are implicated in angiogenesis and fibrogenesis [51]. 

Conflicting results have been obtained regarding the effects of autophagy on LSEC phenotypes and functions. Luo et al. reported that autophagy increases in LSECs in human fibrotic livers [52]. In mouse model of fibrosis, the autophagic degradation of caveolin-1, F-actin remodeling, and the downregulation of the nitric oxide (NO)-dependent pathway occur during the process of CCL_4_-induced LSEC defenestration. The autophagy inhibitor 3-MA could attenuate these effects to maintain LSEC fenestrae and improve liver fibrosis [52]. Ruart et al. reported that the selective disruption of *Atg7* in LSECs aggravates fibrosis by a reduction in intrahepatic NO and impairment in handling oxidative stress, suggesting that autophagy is important for endothelial cell functions during chronic liver injury [53]. 

Luo et al. demonstrated that stress-induced increases in autophagy impair the function of LSECs [52]. In contrast, Ruart et al. suggested that defective autophagy provokes endothelial cell dysfunction [53]. Differences in experimental design (i.e., the use of a chemical inhibitor in rats versus genetic deletion in mice) may explain this discrepancy. Moreover, the interpretation of results differed between the studies. In fact, Ruart et al. also observed an upregulation of endothelial autophagy after 4 weeks of CCL_4_-induced liver fibrosis, identical to the experimental design used by Luo et al. However, further increases in autophagy were not observed when the severity of fibrosis increased (6 weeks), suggesting that autophagy protects LSECs from stress during the early phases of liver injury but is not sufficient to revert damage in advanced stages, leading to endothelial dysfunction. Thus, there are arguments both for and against autophagy inhibition as a therapeutic strategy when considering only the effects on LSECs.

## 4. Conclusions and Future Perspectives

The liver is composed of both parenchymal and nonparenchymal cells with wide variation in phenotypes and functions, and pathological situations may differentially modulate the autophagy status of each cell type [54]. It is not surprising that controversial results have been obtained regarding the impact of autophagy on liver fibrosis, especially when each study highlights the role of a single cell type. Considering that multiple cell types contribute to the complex process of hepatic fibrogenesis [26], it has become apparent that both defective basal levels of autophagy and stress-induced increases in autophagy are equally important in promoting liver fibrosis [55]. 

In a framework that considers this underlying complexity, the therapeutic modulation of autophagy could be a double-edged sword. For example, the blockage of autophagy in mesenchymal cells may attenuate fibrosis [34,35], whereas autophagy is essential to preserve energy homeostasis in other cell types, especially hepatocytes [42]. Thus, therapeutic strategies to inhibit autophagy could be beneficial or detrimental depending on the context. Modulating autophagy for anti-fibrotic therapies needs to be tailored to target specific cell types in the liver. 

### 4.1. Challenges in the Development of Autophagy Modulators

Many challenges still need to be overcome for the clinical application of autophagy-targeting therapeutics for the management of liver fibrosis. First, given the technical difficulties of measuring autophagic flux in humans, our understanding of the role of autophagy in liver fibrosis is based mainly on animal studies. Confirmatory human studies are needed, particularly studies that compare autophagic flux in specific cells of interest directly in normal and fibrotic livers or between cells isolated from normal or fibrotic livers. Second, autophagy-modifying interventions are often given during toxin administration, which prevents not only the differentiation between a true anti-fibrotic effect and the effect of toxin neutralization, but also mimics the clinical application. It is recommended to test autophagy-modifying intervention after the establishment of fibrosis and cessation of the toxin. Finally, as summarized in this review, either increased or defective autophagy could be of importance in liver fibrosis. How can both be taken into account? It is possible that patients may benefit from a sequential approach in which autophagy is first inhibited (or selectively inhibited in fibrogenic cells) and subsequently activated (or selectively activated in parenchymal cells). Future studies are required to establish approaches that account for the dual effects of autophagy in liver fibrosis.

### 4.2. Conclusions

Based on a more comprehensive understanding of the effects of multiple cell types, anti-fibrotic therapy should ideally target neo-angiogenesis, inflammation, and bile ductular proliferation [56]. Autophagy-modifying intervention influences survival in hepatocytes, has anti-angiogenic effects in LSECs, has anti-inflammatory effects in macrophages, and can prevent HSC activation and decrease DR expansion. Thus, targeting autophagy has multiple effects and is a promising therapeutic strategy for the management of liver fibrosis.

## Figures and Tables

**Figure 1 pathogens-09-00225-f001:**
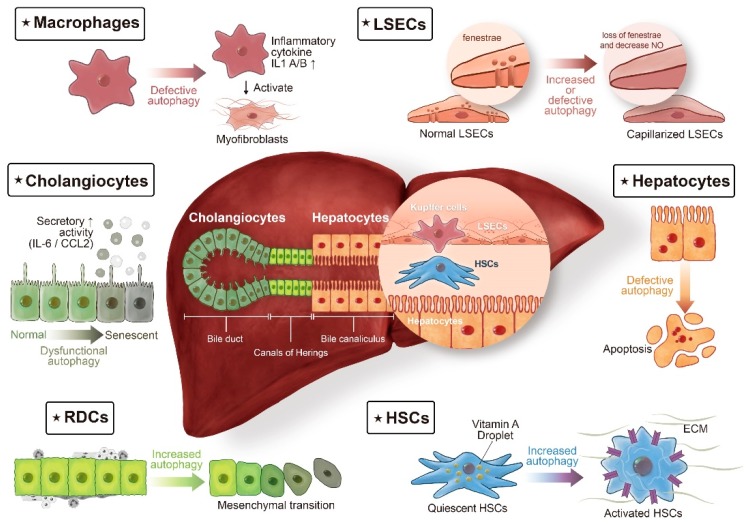
Autophagy in liver fibrosis: different cell populations tell different stories. The liver lobule consists of epithelial cells, hepatocytes, and cholangiocytes, and of non-parenchymal cells, such as macrophages (also termed Kupffer cells), liver sinusoidal endothelial cells (LSECs), and hepatic stellate cells (HSCs). In chronic liver diseases, extension of the fibrotic scars correlates with the presence of ductular reaction (DR). DR refers to extension of reactive ductular cells (RDCs), located in the Canals of Hering, the boundary line between hepatocytes and cholangiocytes. The cellular origin of RDCs is complicated, with different lines of evidence suggesting that in addition to being the progeny of hepatic progenitor cells, RDCs might also be derived from the proliferation of cholangiocytes or from ductular metaplasia of periportal hepatocytes. Autophagy, a complex regulatory pathway in liver fibrosis, exerts its profibrogenic effects in HSCs and RDCs. On the contrary, autophagy maintains cellular homeostasis in hepatocytes, cholangiocytes, macrophages, and LSECs, thereby counteracting fibrogenesis in the liver.

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
