# Peer review of "Complex Cell Type-Specific Roles of Autophagy in Liver Fibrosis and Cirrhosis"

_pathogens, 2020, doi:10.3390/pathogens9030225_

Round 1

Reviewer 1 Report

In this manuscript entitled “complex cell type-specific roles of autophagy in liver fibrosis and cirrhosis”, the authors discus the different cell-types involvement of autophagy in liver pathology. The manuscript is well written. However, some major remarks have to be made.

  • Could the authors clearly state the differences between human and mice when citing scientific work.
  • Please elaborate more extensively on the topic of the reactive ductular cells, macrophages and future prospectives.
  • It would be beneficial to the readers to provide some original pictures of IHC in diseases livers for e.g. LC3 and p62
  • Please remove “also termed biliary epithelial cells” of the subtitle.
  • To aid to reader and substantiate the claims, please reference properly for each specific statement.

Reviewer 2 Report

The review by Hung et al comprehensively reviews the role of autophagy in hepatocytes and other liver parenchymal cells, focusing on the protective aspects of this process as well as how autophagy may lead to liver fibrosis. The review is well laid out and clearly written, and the accompanying figure is very informative. It may be beneficial to expand upon section 2 - reactive ductular cells - and explain in more detail the implications of these observations in the pathogenesis of cirrhosis and fibrosis, especially as the role of HSCs in fibrosis appears to be well characterised. Please describe the assays used - presumably LC3B is an autophagy marker but this is not very clear. 

Author Response

  1. We are grateful to the reviewer for this helpful suggestion. We have added additional description to explain the implications of the results of RDCs in the pathogenesis of cirrhosis. Please see the red highlighted texts in the section 2. We hope that the reviewer will be satisfied with our revision.
  2. Regarding the LC3 assay, in the section of autophagy (lines 8-10 of the paragraph 1), we illustrated that LC3 is an established autophagosome marker and is widely used to monitor autophagy. LC3B is one of the members of the human LC3 gene family. We have added the explanation about the relationship between LC3B and LC3 in the same paragraph. Thank you for your suggestion.

Round 2

Reviewer 1 Report

The authors resolved most of the question properly.

However, some remarks have to be made.

  • It would not be beneficial/allowed to reuse the pictures of previous papers.
    However, since you are describing a general phenomenon I suppose other pictures of a recent study could be taken without much effort?
  • Could the authors describe on higher magnification what they describe as 'ductular reaction". Since it seems now that they are pointing towards "normal" bile ductules, rather than ductular reaction. In addition, in the original paper the authors do not provide a citation of the antibodies (clone, nor product number). Therefore, it is not possible to trace if they used a polyclonal antibody. If so, polyclonal antibodies are known to have an aspecific binding to biliary cells and hepatocytes. 
  • Claims which need proper referencing: line 60-62, line 212-220

Round 3

Reviewer 1 Report

The authors responded to all the questions.

Author Response

Thank you very much for your comment.